# Identification of $CO_2$ adsorption sites on MgO nanosheets by solid-state nuclear magnetic resonance spectroscopy

Jia-Huan Du [1], Lu Chen[2], Bing Zhang[3], Kuizhi Chen[4], Meng Wang[5], Yang Wang[1], Ivan Hung [4], Zhehong Gan[4], Xin-Ping Wu [2], Xue-Qing Gong [2] & Luming Peng [1✉]

The detailed information on the surface structure and binding sites of oxide nanomaterials is crucial to understand the adsorption and catalytic processes and thus the key to develop better materials for related applications. However, experimental methods to reveal this information remain scarce. Here we show that $^{17}O$ solid-state nuclear magnetic resonance (NMR) spectroscopy can be used to identify specific surface sites active for $CO_2$ adsorption on MgO nanosheets. Two 3-coordinated bare surface oxygen sites, resonating at 39 and 42 ppm, are observed, but only the latter is involved in $CO_2$ adsorption. Double resonance NMR and density functional theory (DFT) calculations results prove that the difference between the two species is the close proximity to H, and $CO_2$ does not bind to the oxygen ions with a shorter O⋯H distance of approx. 3.0 Å. Extensions of this approach to explore adsorption processes on other oxide materials can be readily envisaged.

---

[1] Key Laboratory of Mesoscopic Chemistry of Ministry of Education, School of Chemistry and Chemical Engineering, Nanjing University, Nanjing 210023, China. [2] Key Laboratory for Advanced Materials, Centre for Computational Chemistry and Research Institute of Industrial Catalysis, East China University of Science and Technology, 130 Meilong Road, Shanghai 200237, China. [3] Lam Research Corporation, Fremont, California, CA 94538, USA. [4] National High Magnetic Field Laboratory, 1800 East Paul Dirac Drive, Tallahassee, FL 32310–3706, USA. [5] College of Chemistry and Molecular Engineering (CCME), Peking University, Beijing 100871, China. ✉email: luming@nju.edu.cn

Metal oxides play an essential role in many fields of chemistry, physics, material science, biology, and environmental science[1–3]. Their corresponding nanomaterials, which have a high surface area, often exhibit unique or improved properties, especially for applications in gas sensing, gas storage, and catalysis[4–6]. Identifying gas adsorption sites in oxide nanomaterials and gas-surface interactions are crucial in designing and tailoring materials with improved performances. A variety of characterization methods, including electron microscopy and spectroscopic techniques, have been used to explore this issue, which is challenging due to the complex nature of the surface[7]. Despite the very high-resolution microscopy can possibly achieve these days, the images may not be representative of the whole structure and therefore quantitative analysis cannot be performed[8]. Spectroscopic investigations, such as IR, often require probe molecules, e.g., CO, making such methods indirect[9] and unsuitable for in situ studies, in which probe molecules may interfere. Furthermore, many of these techniques require ultra-high or high vacuum conditions[10,11], thus the information obtained may not correspond to the real application. Therefore, it is required to develop alternative approaches to understand the gas binding sites in oxide-based materials.

Recently, we showed that $^{17}$O solid-state NMR spectroscopy provides very high resolution and oxygen species at different facets or different layers in oxide nanostructures can be clearly distinguished based on $^{17}$O NMR shifts[12–15]. This approach does not introduce any external species that may potentially alter the structure or properties of oxide surface, such as probe molecules[12]. Therefore, it provides unprecedented opportunities to explore the specific gas adsorption sites and the interactions on the surface for metal oxides.

MgO, a common oxide with a rock-salt structure, is of fundamental importance in physics[16,17], chemistry[18–20], nanotechnology[21], as well as earth and space science[22–24]. In particular, chemisorption using MgO represents a promising strategy for reducing the levels of atmospheric $CO_2$, the major greenhouse gas causing rapid and adverse climate change[25]. This strategy has attracted a lot of research attention owing to its wide operation temperature, low cost, and insusceptibility to humidity[25–27]. For example, MgO was recently spread over land for direct $CO_2$ capture and demonstrated its scalable potential to remove more than 2 Gt $CO_2$ per year at a low cost[28]. Here, we demonstrate that $^{17}$O solid-state NMR spectroscopy can be used as a primary method to study details of $CO_2$ adsorption on MgO nanosheets. The approach, which is potentially extendable to other oxides, can distinguish two surface oxygen species only slightly different on the third coordination shell, identify the sorption sites of $CO_2$, and reveal the nature of these sites as well as the reasons behind the differences in adsorption properties.

## Results

Previous studies have shown that the surface structure and $CO_2$ adsorption capacity of MgO vary significantly after calcining at a temperature higher than 800 K[29,30]. Thus, two different temperatures of 773 and 1073 K were used to prepare MgO nanomaterials. The powder X-ray diffraction (XRD) patterns of the MgO materials calcined at both temperatures are indexed and periclase MgO structure is confirmed (PDF No.71-1176) (Supplementary Fig. 1). High-resolution transmission electron microscopy (HRTEM) images show the nanosheet morphology of the MgO materials and the nanosheets are denoted as NS-773 and NS-1073 (NS for nanosheets), according to the corresponding thermal treatment temperatures (Supplementary Fig. 2). The primary exposed facets are determined as (111), and the thicknesses of both nanosheet samples are approx. 10 nm, which

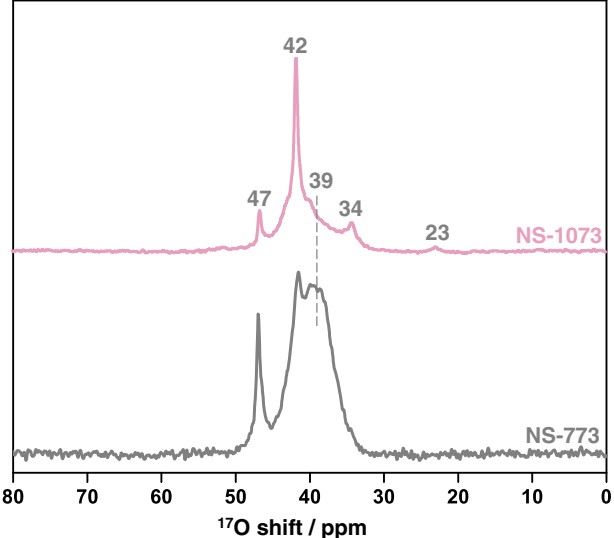

**Fig. 1 $^{17}$O single-pulse MAS NMR spectra of MgO nanosheets.** External field: 19.6 T; MAS rate: 16 kHz; recycle delay: 5 s. The spectra are plotted such that the highest peak in each spectrum is at the same height and the relative intensities of different peaks can be conveniently compared.

agrees with the average crystal sizes of MgO nanosheets calculated based on the XRD data using the Debye–Scherrer equation.

MgO nanosheets were then mixed with $^{17}$O$_2$ and heated at 623 K for $^{17}$O enrichment. First, $^{17}$O single-pulse MAS NMR was used to study NS-773 and NS-1073 (Fig. 1). For the data obtained at an external field of 19.6 T, the well-resolved sharp peak at 47 ppm found in both spectra can be ascribed to 6-coordinated oxygen ions ($O_{6C}$) in the bulk part of the MgO nanosheets[31,32]. Lower frequency resonances are also observed, including a relatively sharp and intense signal at 42 ppm, a broader component centered at around 39 ppm along with weaker peaks at approx. 34 and 23 ppm. Since $^{17}$O is a quadrupolar nucleus, peaks or discontinuities may not necessarily mean different sites and they can just arise from non-zero second-order quadrupolar interactions. Therefore, NMR spectra were also obtained for both samples at 9.4 T (Supplementary Fig. 3b). The linewidths of the peaks owing to large quadrupolar couplings become much wider at a lower external magnetic field. The fact that the data at 9.4 T do not show significant line broadening, indicates that these peaks are due to oxygen species in different chemical environments, which are associated with small quadrupolar interactions. An $^{17}$O NMR resonance centered at 41 ppm was observed in $^{17}$O-enriched nanophase MgO by Chadwick et al., which was assigned to oxygen sites perturbed by nearby H or CH$_3$ at the third-nearest-neighbor on the surface[33]. Therefore, the peaks at 42 and 39 ppm in our data are tentatively assigned to surface oxygen species. $^{17}$O enrichment of oxides with $^{17}$O$_2$ gas usually requires high temperature and most $^{17}$O isotopes diffuse into the bulk part of the sample, leading to poor sensitivity of $^{17}$O NMR observations of surface species. In our $^{17}$O NMR data, however, the intensities of the resonances for $O_{6C}$ in the bulk are only a small fraction of the total spectral intensity (~10%), indicating that a large fraction of the signal comes from the surface of the nanosheets, which is probably owing to the relative low enrichment temperature and/or the high surface area of the MgO nanosheets (Supplementary Fig. 3). The small fraction of the 'bulk' signal is also related to the relatively short recycle delay used, which is long enough for quantitative measurements of surface oxygen species yet not enough for bulk, because the longitudinal relaxation time ($T_1$) is shorter for the surface than the bulk (Supplementary Fig. 4)[14].

Comparing the NMR data of NS-1073 and NS-773, the peak centered at 39 ppm has the strongest intensity for NS-773, while this resonance is much weaker and the peak at 42 ppm dominates in the spectrum of NS-1073, indicating that the concentrations of the two corresponding surface species vary when the MgO nanosheets are thermally treated at different temperatures.

The $^{17}O$-enriched MgO nanosheets were activated under vacuum and then dosed with $^{13}CO_2$ to explore the adsorption capacity and the surface sites involved in $CO_2$ adsorption. The $^{13}C$ MAS NMR spectrum of NS-773 shows a single peak at 167 ppm, which can be ascribed to unidentate carbonate species[34,35], suggesting that $CO_2$ was chemisorbed on the surface of MgO nanosheets (Fig. 2a and Supplementary Fig. 5). Since the experiments were performed at a relatively low temperature of 313 K, the carbonate is formed mostly on the surface[36,37]. The $^{13}C$ NMR spectrum of NS-1073 is similar and the peak maximum is observed at a slightly lower frequency of 166 ppm. $CO_2$ adsorption data (Supplementary Fig. 6) and quantitative analysis on the $^{13}C$ NMR spectral intensities show that the $CO_2$ adsorption capacity (per mass) is around 55% higher for NS-1073 than NS-773, despite the fact that the former showed a much smaller surface area (142 $m^2 \cdot g^{-1}$) than the latter (227 $m^2 \cdot g^{-1}$) (Supplementary Fig. 7), which is similar to previous observations[30]. Such differences probably arise from the different relative concentrations of different surface species in the two samples.

To identify the surface oxygen species involved in $CO_2$ adsorption, $^{17}O$ MAS NMR spectroscopy was applied to study the changes on the surface of MgO nanosheets. For NS-773, although both the peak at 42 ppm and the shoulder at 39 ppm have a very strong intensity in the spectrum before adsorption, only the intensity of the peak at 39 ppm remains after adsorption while the resonance at 42 ppm becomes much weaker (Fig. 2b). The difference spectrum clearly shows that the surface species giving rise to the peak at 42 ppm are likely involved exclusively in binding $CO_2$, converting to surface carbonate species upon adsorption, as suggested by $^{13}C$ NMR results. Carbonates oxygens have a $^{17}O$ chemical shift of 223–274 ppm and a large quadrupolar coupling constant ($C_Q$) of approx. 7 MHz, according to previous $^{17}O$ NMR calculations on $MgCO_3$ clusters[38] and experimental observations of $CaCO_3$[39]. However, such a signal is not observed in our data, presumably due to the large linewidth arising from the large second-order quadrupolar interaction. Similar results were obtained for NS-1073 (Fig. 2c). Although the difference spectrum also shows weak signals at 52, 34, and 23 ppm, which may arise from a small amount of different surface sites, the majority of the intensity comes from the peak at 42 ppm, implying that the corresponding species are the key to $CO_2$ adsorption for MgO nanosheets. Therefore, only the peaks at 39 and 42 ppm are focused on in the following study.

Since the peaks at around 41 ppm may be related to the surface H or $CH_3$ according to previous studies[33], $^1H$ NMR spectroscopy was used to study the H species in MgO nanosheets (Supplementary Fig. 8). The overlapping resonances observed in the spectra of NS-773 and NS-1073 centered at around $-1$ to $-0.5$ ppm can be ascribed to the hydrogen-bond acceptor and/or isolated hydroxyl groups[40–42]. Quantitative measurements show that the total $^1H$ NMR intensity of NS-1073 is only around 20% of NS-773, indicating significant H loss during thermal treatment. While the $CH_3$ proton is not resolved in the $^1H$ NMR spectrum, neither $^{13}C$ single pulse (Supplementary Fig. 9) nor $^1H \rightarrow {}^{13}C$ cross-polarization (CP) MAS NMR (Supplementary Fig. 10) spectrum of NS-773 exhibits the signal for $CH_3$, implying the proposed $CH_3$ is not present in our MgO nanosheets.

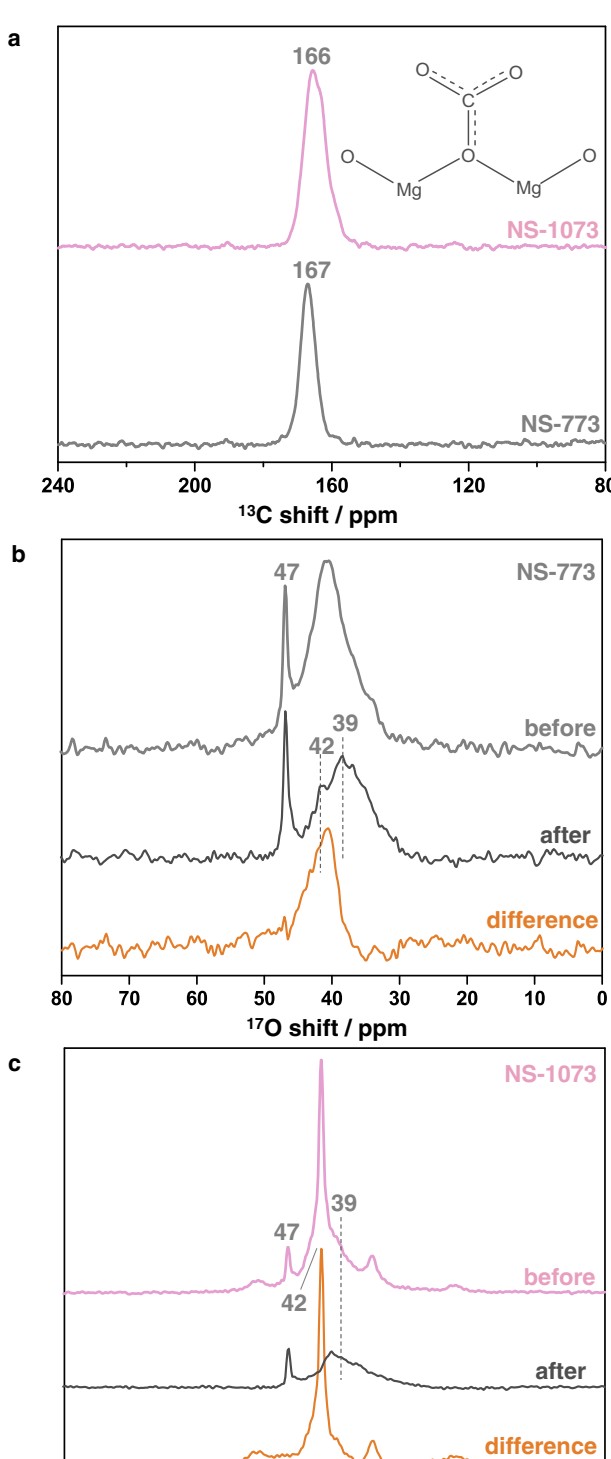

**Fig. 2 $^{13}C$ and $^{17}O$ single-pulse MAS NMR data of MgO nanosheets adsorbed with $CO_2$. a** Single-pulse $^{13}C$ MAS NMR spectra of NS-773 and NS-1073 after $^{13}CO_2$ chemisorption at 9.4 T. MAS rate: 16 kHz; recycle delay: 40 s. $^{17}O$ NMR spectra of NS-773 (**b**) and NS-1073 (**c**) before and after $^{13}CO_2$ chemisorption along with the difference spectrum obtained at 9.4 T. MAS rate: 14 kHz; recycle delay: 5 s.

Despite strong evidence for the presence of hydroxyl sites in MgO nanosheets, no peak attributed to these species appear in the single-pulse $^{17}O$ NMR data (Fig. 1 and Supplementary Fig. 11). The hydroxyl groups are expected to have a similar chemical environment as oxygen ions in $Mg(OH)_2$, and thus an isotropic chemical shift of around 20 ppm and a very large $C_Q$ of approx. 6.8 MHz (obtained on $Mg(OH)_2$)[43]. The absence of such signals is presumably due to the relatively low concentration of H in MgO nanosheets as well as the wide linewidth. On the other hand, broad resonances at low frequencies (~20 to −250 ppm) appear in the spectra of the MgO nanosheets exposed to water vapor, indicating hydroxyl species can be observed at a higher concentration due to large quadrupolar interaction (see Supplementary Figs. 12, 13 and related discussion).

$^{17}O$-$^1H$ REDOR NMR spectroscopy, which probes the internuclear proximity between oxygen and proton, was further applied to explore the nature of the resonances at 42 and 39 ppm (Fig. 3)[44]. Similar intensities in both the control and double resonance spectra are observed for the sharp resonance at 47 ppm, even at a very long recoupling time of 2.75 ms, because oxygen ions in the bulk part are far away from any proton, which is on the surface (Fig. 3a). In contrast, both the peaks at 42 and 39 ppm show lower intensity at a long recoupling time in the double resonance spectrum compared to the control spectrum, indicating that these oxygen species are near protons and thus are indeed on the surface. Nonetheless, the relatively small REDOR fractions ($\Delta S/S_0$, where $\Delta S$ and $S_0$ are the intensity of the difference spectrum and the control spectrum, respectively) with short recoupling time suggest these oxygen sites are not directly bound to proton (Fig. 3b and Supplementary Fig. 14)[14,15,45,46]. The difference spectrum clearly shows that the signal at 39 ppm is associated with a greater REDOR fraction than the peak at 42 ppm, suggesting the oxygen ions in the former environment are closer to the proton.

The distances between O and H were extracted by measuring the REDOR fractions as a function of recoupling time and performing numerical simulations[47]. Based on a model containing a single O–H spin pair, O–H distances of ~3.7 Å and 3.0 Å are determined for the surface oxygen ions giving rise to the peak at 42 and 39 ppm, respectively, the latter consistent with the distance between an oxygen ion and a proton in its third coordination shell (3.2 Å, see Fig. 3b). A longer O–H distance for the peak at 42 ppm suggests that this type of oxygen ion is less affected by the closest proton, which is probably in the fifth coordination shell or further away. Therefore, considering the observation that the $^1H$ NMR intensity of NS-773 is much stronger than NS-1073, and the fact that the peak at 39 ppm dominates the $^{17}O$ NMR spectrum of NS-773 while the resonance at 42 ppm is the strongest signal for NS-1073, the $^{17}O$ NMR peaks at 39 and 42 ppm are assigned to surface 3-coordinated oxygen ions ($O_{3C}$) with and without proton in the third coordination shell on the MgO (111) facets (Fig. 3b). Since $CO_2$ only binds to the species giving rise to the peak at 42 ppm, the absence of proton nearby is crucial for successful $CO_2$ adsorption. It can also explain the stronger spectral intensity for the peak at 39 ppm compared to the hydroxyl species (not observed in $^{17}O$ NMR) because one proton can generate up to six bare surface oxygen ions in the surrounding giving rise to the resonance at 39 ppm (Supplementary Fig. 15).

The above results are also in good agreement with UV-Vis diffuse reflectance spectra of MgO nanosheets (Supplementary Fig. 16). Before $CO_2$ adsorption, only one intense adsorption band at ~280 nm (4.4 eV) is observed for both samples, which can be attributed to the excitation of $O_{3C}$[48–50]. The intensity of this band decreases significantly after $CO_2$ adsorption, while the intensity of the band corresponding to $O_{4C}$ sites (230 nm or 5.4 eV) increases, indicating $O_{3C}$ species are converted to $O_{4C}$ by forming unidentate carbonate species during the process (Fig. 2).

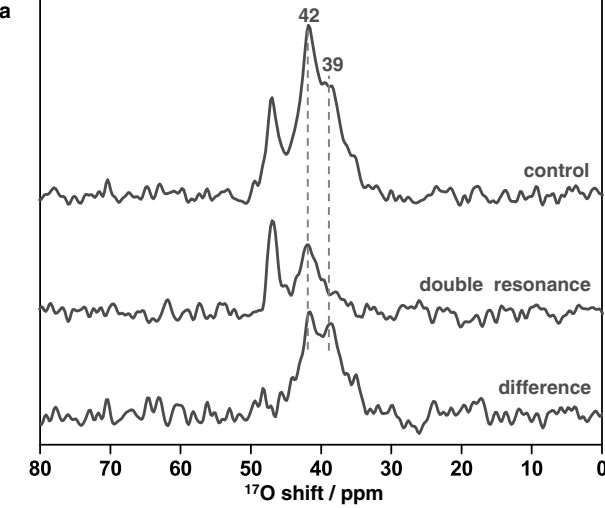

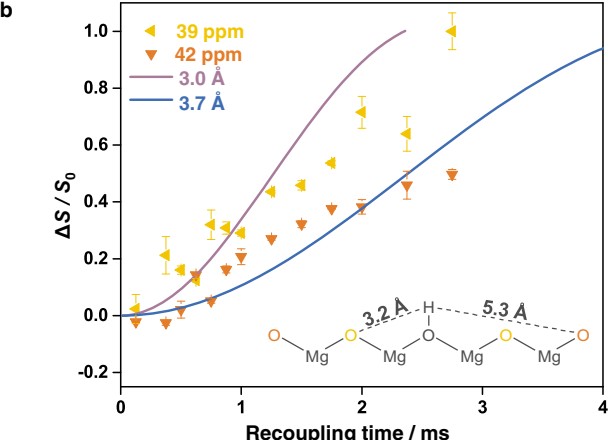

**Fig. 3 $^{17}O$-$^1H$ REDOR NMR data of MgO nanosheets (NS-773). a** REDOR NMR spectra with a recoupling time of 2.75 ms. **b** REDOR fraction ($\Delta S/S_0$) as a function of recoupling time with the schematic surface structure of MgO. The yellow and orange triangles represent 39 and 42 ppm, respectively. External field: 18.8 T; MAS rate: 16 kHz; recycle delay: 5 s.

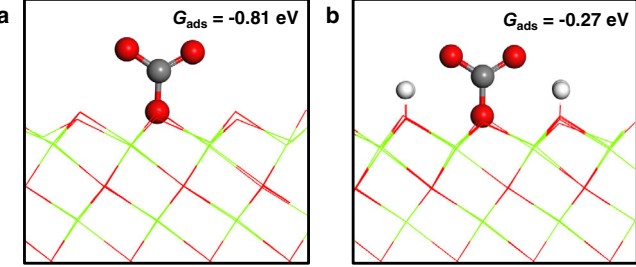

**Fig. 4 Calculated structures of $CO_2$ adsorption on MgO.** $CO_2$ unidentate adsorption on pristine MgO(111) (**a**) and MgO(111)-4H (**b**). Hydrogen, oxygen, carbon, and magnesium atoms are in white, red, gray, and green, respectively. Oxygen and magnesium atoms uncoordinated to $CO_2$ are displayed with lines.

Finally, density functional theory (DFT) calculations were performed to calculate the free energies of $CO_2$ adsorption ($G_{ads}$) on pristine MgO(111) surface and hydrogen pre-adsorbed MgO(111) surface (denoted as MgO(111)-4H), which correspond to the adsorption on the surface oxygen ion without and with a proton at the third coordination shell, respectively (Fig. 4). The calculated $G_{ads}$

for unidentate adsorption of $CO_2$ on pristine MgO(111) is −0.81 eV, indicating strong $CO_2$ adsorption on this surface. In contrast, the calculated $G_{ads}$ is much closer to zero (−0.27 eV) for MgO(111)-4H, suggesting that $CO_2$ molecules are much less inclined to be adsorbed at these sites. This result, in combination with the NMR data, confirms that protons can poison nearby bare oxygen species at the MgO(111) facets and inhibit $CO_2$ adsorption on these sites. Thus, a decrease in the proton concentration can enhance the $CO_2$ adsorption capacity significantly.

## Discussion

With isotopic labeling $^{17}O$ in MgO nanomaterials, the surface structure and $CO_2$-surface interaction can be conveniently probed with $^{17}O$ solid-state NMR spectroscopy. We show that bare surface oxygen sites on MgO nanosheets different only on the third coordination shell are distinguished according to $^{17}O$ NMR shifts (42 and 39 ppm), and the oxygen species responsible for $CO_2$ adsorption (42 ppm) are clearly identified based on the spectral intensity change before and after adsorption. Double resonance NMR techniques and DFT calculations show that $CO_2$ adsorption on the surface oxygen atom is strongly inhibited by H species at its third coordination shell and the results shed light on the variations in the $CO_2$ adsorption capacity upon thermal treatment.

Unlike the previous probe-molecule-assisted NMR method[51–53], in which the surface structural information is observed indirectly through adsorbed molecules, our $^{17}O$ NMR-based approach focuses on the solid surface itself and can follow its evolution upon sorption directly at high resolution. This $^{17}O$ NMR method may also be combined with signal enhancement techniques, such as dynamic nuclear polarization[54–58], to study materials with larger sizes and smaller surface area, and thus potentially be extended to a variety of other metal oxides to investigate gas adsorption processes and help to design related materials with improved properties for adsorption and catalysis. Despite these possible promising applications, it is challenging to detect species with a large quadrupolar interaction at a very low concentration by using this approach. Although oxygen ions in most metal oxides do have a small to medium $C_Q$ due to the small electronegativity values of common metals[59], large quadrupolar interactions may be found in oxygen atoms bound to elements with a relatively large electronegativity, such as hydroxyl groups in this case, or in surface sites with lower coordination numbers and/or dangling bonds[14]. Obtaining data at a higher external magnetic field, which decreases the line broadening due to quadrupolar interactions, may alleviate the problem.

## Methods

**Synthesis of MgO nanosheets**. MgO nanosheets were prepared by a modified aero-gel method[60]. In a typical synthesis, 1.0 g Mg belt was polished by using sandpaper and cleaned with acetone before it was dissolved in methanol (~44 mL), forming a 10 wt % $Mg(OCH_3)_2$ solution in methanol. Then 4-methoxybenzyl alcohol (BZ) was added to the above solution (with a ratio of Mg:BZ = 2:1). After the mixture was stirred for 5 h, 30 mL water-methanol solution (molar ratio of $H_2O$ to Mg is 2:1) was added dropwise. Then the mixture was further stirred for 12 h before it was transferred to an autoclave. After purging the autoclave with Ar for 10 min, the pressure was raised to 10 bar with Ar. Then the autoclave was heated to 538 K and kept at the temperature for 15 h before the vapor in the autoclave was released in 1 min. The resulting white power was heated to 773 K with a ramping step of 3 K/min, and it was kept at the temperature for 6 h to obtain the white powder of MgO nanosheets exposing mostly {111} facets used in this work.

**$^{17}O$-Enrichment procedures and $CO_2$ chemisorption**. MgO nanosheets were first heated at elevated temperature (773 or 1073 K) under vacuum for 2 h before they were cooled to room temperature. After that, 90% $^{17}O$-enriched $O_2$ gas was introduced to the samples through a vacuum line. In order to selectively label mostly the surface oxygen species, the mixture was heated to 623 K and kept at the temperature for 14 h. For $CO_2$ chemisorption, $^{17}O$-labeled MgO nanosheets were first exposed to vacuum and then 170 mbar $^{13}CO_2$ at 313 K for 6 h. After that, the samples were exposed to vacuum again to remove physisorbed $CO_2$.

**Characterization**. The X-ray powder diffraction (XRD) measurements were carried out using a SHIMADZU XRD-6000 diffractometer with a Cu Kα source (λ = 1.54178 Å) operating at 40 kV and 40 mA, in the 2θ range of 20−80°. High-resolution transmission electron microscopy (HRTEM) images were acquired on a JEOL JEM-2100 instrument with an acceleration voltage of 200 kV. The Brunauer −Emmett−Teller (BET) specific surface area was obtained by nitrogen adsorption-desorption at 77 K with a Micromeritics Tristar 2020 apparatus. The UV-Vis measurements were operated using a Shimadzu UV-3600 at room temperature (200–800 nm). $^{17}O$ MAS NMR spectra obtained at 19.6 T were recorded with a Bruker Avance NEO spectrometer with 3.2 mm MAS probes. Excitation pulses with a width of 2 μs, corresponding to a solution π/6 flip angle, were used. $^{17}O$-$^{1}H$ REDOR NMR spectra acquired at 18.8 T were recorded with a Bruker Avance III spectrometer with a double-tuned 3.2 mm MAS probe. A direct excitation pulse sequence was used $^{17}O$ pulses of 5 and 10 μs (solution π/2 and π pulses) and $^{1}H$ π pulses of 6.7 μs for recoupling. O–H distances were extracted assuming there is only a single O–H spin pair[47]. $^{17}O$, $^{1}H$, and $^{13}C$ MAS NMR data obtained at 9.4 T were recorded with a Bruker Avance III spectrometer with 3.2 and 4.0 mm MAS probes. Excitation pulses with a width of 1.2 μs, corresponding to a solution π/6 flip angle, were used for single-pulse $^{17}O$ NMR experiments. π/2 pulses with a length of 2.1 and 2.9 μs were applied, for single-pulse $^{1}H$ and $^{13}C$ MAS NMR, respectively. $^{17}O$, $^{1}H$, and $^{13}C$ chemical shifts were referenced to $H_2O$ at 0.0 ppm, adamantane at 1.92, and 38.5 ($CH_2$) ppm, respectively. All the samples were packed in an $N_2$ glove box.

## Data availability

All data supporting the findings presented here is included in the manuscript, its supporting information, or from the authors upon request.

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

## Acknowledgements

This work was supported by the National Natural Science Foundation of China (NSFC) (91745202 and 21972066), NSFC—Royal Society Joint Program (21661130149). L.P. thanks the Royal Society and Newton Fund for a Royal Society—Newton Advanced Fellowship. The ECUST group thanks to the Programme of Introducing Talents of Discipline to Universities (B16017). A portion of this work was performed at the National High Magnetic Field Laboratory, which is supported by the National Science Foundation Cooperative Agreement No. DMR-1644779 and the state of Florida.

## Author contributions

J.-H.D. and L.P. supervised the project. J.-H.D., B.Z., K.C., M.W., Y.W., I.H., and Z.G. designed and carried out the experiments. L.C., X.-P.W., and X.-Q.G. carried out the DFT calculations. J.-H.D. and L.P wrote the paper; and all authors participated in the analysis of the data and discussions of the results, as well as preparation of the paper.

## Competing interests

The authors declare no competing interests.
