## [Peer Review File · Nature Communications]

Identification of CO₂ adsorption sites on MgO nanosheets by solid-state nuclear magnetic resonance spectroscopyREVIEWER COMMENTS

Reviewer #1 (Remarks to the Author):

Du and coauthors present a structural study of oxide nanomaterials using advanced methods in solid-state NMR spectroscopy in order to identify at the surface of MgO nanomaterials, those sites that are primarily involved into the adsorption of gas. Here they use CO₂ adsorption as a model reaction for further extrapolation of their NMR protocol to other gas adsorption (not investigated in the present paper). The authors focus on the use of ¹⁷O NMR as a sensitive probe, a nucleus that has gained large interest through the last two decades especially with very high magnetic fields now readily available. The present study is first based on the identification and assignment of two resonances at 39 and 42 ppm attributed to surface oxygen sites. Further, the authors use a more advanced NMR technique, the ¹H-¹⁷O REDOR NMR pulse sequence, to determine proton-oxygen distance and thus, propose a model for the surface of their MgO nanomaterials where the resonance at 39 ppm would be associated to hydroxyl oxygen. Previously, by investigating two MgO materials, calcined at two temperatures (773 and 1073 K), Du et al. showed that the oxygen site at 42 ppm is the one most affected by the CO₂ adsorption.

I have a few questions about the methods and protocol.

- The authors consider a model with an isolated hydroxyl group in order to interpret their REDOR curves. Would it be more realistic to consider a multispin system (rather than a spin pair) to explain the REDOR curves especially at long recoupling time (> 1.5 ms)?
- The ¹⁷O enrichment is clearly selective and assumed by the authors in order to reduce enrichment of the oxygen in the bulk only. Moreover, they take advantage of the fact that the targeted surface oxygen sites (at 39 and 42 ppm) present small quadrupolar couplings. On the contrary, they do not observe the Mg(OH)₂ sites. This is bothersome. I understand the difficulty to observe such signal with large second-order quadrupolar couplings. Would ¹H-¹⁷O CP be an option to highlight those sites?

The data and their interpretations represent comprehensive and coherent work. The text is clear with very few (typo) errors and the figures are of high quality. However, if the data are definitely of interest for NMR spectroscopists and to a lesser extent to chemists interested in characterization methods, it seems to me, on the other hand that these results do not fulfill the selection criteria for a publication to Nature Communications, as requested by your editorial board : "...journal aim to represent important advances of significance to specialists within each field". The NMR methods used here are not new, their application to characterize gas adsorption is interesting but lack a broader application (another gas than CO₂ for example).

In my opinion, I would not recommend the publication of this manuscript in Nature Communications; I would rather suggest the submission of a full article (including data from the Supp Info file) in a more specialized journal. I leave that final decision to the editor.

Reviewer #2 (Remarks to the Author):

It is good to see a paper with a very straightforward goal that if successful could make a big difference into understanding an important issue. The central premise is that a spectroscopic approach that can elucidate the coordination of molecules on oxide surfaces without the need for a probe molecule or an indirect model could present significant advantages. Here ¹⁷O solid-state NMR of MgO surfaces of ¹⁷O isotopically-enriched MgO nanosheets is used to investigate the approach. Although this is a model example, the principle looks as though it is well demonstrated. Through different preparations and looking at the effect of introducing CO₂ the different surfaces sites are plausibly identified and the preferential interaction with CO₂ determined. A very clear prediction is made in that the surface OHs present inhibit several of the surrounding surface oxygen sites from coordinating CO₂.

The use of three magnetic fields for ¹⁷O (effectively two for straight spectral comparison – 9.4 and 19.6 T) as well as carrying out REDOR, along with observing ¹H and ¹³C a high quality NMR

methodology is used. There is the back up of other characterisation techniques. There is a good split of the data presented directly in the paper and that given in the Supplementary Information (SI). The only substantive scientific comments are for perhaps some minor expansion of the explanation of some of the technical NMR points given that Nat. Comm. is not a specialist NMR journal some unpacking of the meaning of observation of the relative linewidth at 9.4 and 19.6 T could be added (only a few sentences). Also some of the spectra appear to show some weak spinning sidebands and these could be usefully marked (e.g. Fig. 1 NS-1073 at around 23 ppm?)

The writing for the most part is very clear so that it is easy to follow the arguments developed. The experiments are clearly described with a judicious use of figures split between the main paper and the SI. A very sensible balance is struck in the referencing which shows a good appreciation. The conclusion one comes to is this is a high quality piece of research that has a good deal of novelty. The issue of being able to elucidate oxide surface coordination sites directly using ^{17}O NMR could have a very significant impact. There are several disciplines this approach would be applicable to. Hence on balance I see there are good reasons for publication in Nat. Comm. There are some minor points that need tidying and these are listed below.

Minor corrections

1. p3, line 7, it would be more usual to refer to microscopy than microscopic
2. p3, line 16, approaches is better
3. p3, line 18, should be showed
4. p10, line 1, delete s to give evidence
5. p12, line 1 after the figure caption, should be fractions not factions
6. p16, line 9, probably would be better just to insert ^{17}O NMR before method
7. p17, line 5, should be powder

Reviewer #3 (Remarks to the Author):

Du et al present an interesting study on the use of ^{17}O enrichment and ^{17}O NMR to understand catalytic behavior in MgO nanosheets. Using ^{17}O NMR and DFT, the authors show that proton poisoning inhibits CO_2 adsorption on nearby surface oxygen sites on MgO. The work has been carefully conducted and provides an advanced understanding of surface chemical reactions without the need for UHV or probe molecules. However, there are several areas that could use additional explanation and analysis, detailed in the comments below:

1. On page 6, the authors assign the ^{17}O NMR spectra to bulk O in MgO at 47 ppm, and the remaining resonances to surface sites. It would be helpful if the authors could be more quantitative with the data (i.e., what is the % MgO bulk in the two samples shown in Fig 1?). They note that the T1 of the surface vs bulk is very different, based on Figure S4, but do not state the T1 of the bulk? Can the authors comment on this value and whether or not quantitative NMR is possible? They also go on to describe changes in the relative intensity of bulk vs surface between the two samples, but it's not clear if this is due to O_2 diffusion into the bulk at higher annealing temperatures or an actual change in the surface chemistry.
2. Figure 1: should the resonances at 34 and ~25 ppm be labeled? The peak at 25 ppm is not mentioned in the text.
3. On page 9, the authors note that the resonance corresponding to adsorbed carbonate is not observed in ^{17}O NMR possibly due to the large linewidth from the second order quadrupolar interaction. If true, it would be interesting to analyze the spectra collected at 18.8 T (e.g. from the

REDOR experiments in Fig 3) in more detail or simulate the spectral broadening expected from these sites to discern whether they can be observed.

4. Also on page 9, the authors rationalize that CH₃ groups are not present based on ¹H NMR and ¹H-¹³C CPMAS NMR due to the lack of signal. It would be useful to show the full ¹³C NMR spectra in the SI to show the lack of CH₃ groups, as only the carbonate range appears in Fig 2 (similar to how they show an expanded view of the ¹⁷O NMR).

5. The authors should include error bars for the REDOR measurements reported in Fig 3. The suggested binding sites on the surface are also not entirely consistent with the proposed structures (5.3 Å vs 3.7 Å). Are there other possible surface structures that are not accounted for?

Response to Reviewers

Reviewer #1 (Remarks to the Author):

Du and coauthors present a structural study of oxide nanomaterials using advanced methods in solid-state NMR spectroscopy in order to identify at the surface of MgO nanomaterials, those sites that are primarily involved into the adsorption of gas. Here they use CO₂ adsorption as a model reaction for further extrapolation of their NMR protocol to other gas adsorption (not investigated in the present paper). The authors focus on the use of ¹⁷O NMR as a sensitive probe, a nucleus that has gained large interest through the last two decades especially with very high magnetic fields now readily available. The present study is first based on the identification and assignment of two resonances at 39 and 42 ppm attributed to surface oxygen sites. Further, the authors use a more advanced NMR technique, the ¹H-¹⁷O REDOR NMR pulse sequence, to determine proton-oxygen distance and thus, propose a model for the surface of their MgO nanomaterials where the resonance at 39 ppm would be associated

to hydroxyl oxygen. Previously, by investigating two MgO materials, calcined at two temperatures (773 and 1073 K), Du et al. showed that the oxygen site at 42 ppm is the one most affected by the CO₂ adsorption.

I have a few questions about the methods and protocol.

- The authors consider a model with an isolated hydroxyl group in order to interpret their REDOR curves. Would it be more realistic to consider a multispin system (rather than a spin pair) to explain the REDOR curves especially at long recoupling time (> 1.5 ms)?

Response: We thank the reviewer for the helpful suggestion. Of course, a simulation considering more than two spins should be better. As requested, we tried to use SIMPSON software (M. Bak, J. T. Rasmussen, N. C. Nielsen, SIMPSON: A General Simulation Program for Solid-State NMR Spectroscopy, J. Magn. Reson., 2000, 147, 296) to simulate the REDOR curves for one oxygen atom with one or more hydrogen

atoms around. As can be imagined, simulations considering such multi-spin system can be quite challenging and the simulation is expected to take much longer time with each additional spin, which was also explained in the literature (J. Magn. Reson. 2006, 178, 248.). We managed to do simulations with up to 3 hydrogen atoms in the fifth coordination shell of one oxygen atom (Fig. R1 and R2, and the SIMPSON simulation input files are shown below. The simulation considering 3 hydrogen atoms already took several days to finish and the simulation considering 4 hydrogen atoms failed). It is clear that with more hydrogen atoms around, the REDOR fraction increases steadily, indicating that the effects from each additional hydrogen atom add up (spheres in Fig. R2). A clear trend can be seen that a higher REDOR fraction can be expected with 4 or more hydrogen atoms. From trend of the obtained simulation results, it is expected that one oxygen atom with around 6 hydrogen atoms at the fifth coordination shell should generate a REDOR curve similar to the experimental data for the peak at 42 ppm in NS-773. Because the simulations considering more than 3 hydrogen atoms failed, we decided to leave the related results in the response to reviewers, which is still available to the readers.

Fig. R1. The structure for SIMPSON simulation for the center oxygen ion (marked with a blue triangle) with one (A), two (B), or three (C) H atoms. The oxygen ions in the third and fifth (or more) coordination shell of H (white) are shown in yellow and orange, respectively.

Fig. R2. REDOR fractions obtained with SIMPSON simulations considering structures shown in Fig. R1, in comparison to experimental data and analysis shown in Fig. 3.

Simpson simulation input file (Oxygen ion with one hydrogen atom: (1O + 1H)):

```
spinsys {
  channels 17O 1H
  nuclei 17O 1H
  dipole 1 2 113 0 0 0
  shift 1 3p 10p 0.5 50 20 10
  quadrupole 1 2 100 0 0 0
}

par {
  variable index 1

  np 200
  spin_rate 16000
  proton_frequency 800e6
  start_operator 11x
  detect_operator 11p
  method direct
  crystal_file rep320
  gamma_angles 18
  sw spin_rate/2
  variable tsw 1e6/sw
  verbose 1101
  variable rfF1 50000
  variable rfF2 74600
  variable t180F1 0.5e6/rfF1
  variable t180F2 0.5e6/rfF2
  variable tr1 0.5e6/spin_rate-0.5*t180F1-0.5*t180F2
  variable tr2 0.5e6/spin_rate-t180F2
}

proc pulseseq {} {
  global par

  reset
```

```

delay $par(tr2)
pulse $par(t180F2) 0 x $par(rfF2) x
delay $par(tr2)
pulse $par(t180F2) 0 x $par(rfF2) y
store 1

reset
acq
delay $par(tr2)
pulse $par(t180F2) 0 x $par(rfF2) x
delay $par(tr1)
pulse $par(t180F1) $par(rfF1) x 0 x
delay $par(tr1)
pulse $par(t180F2) 0 x $par(rfF2) x
delay $par(tr2)
pulse $par(t180F2) 0 x $par(rfF2) y
store 2
acq

for {set i 2} {$i < $par(np)} {incr i} {
  reset
  prop 1
  prop 2
  prop 1
  store 2
  acq
}
}
proc main {} {
  global par

  set f [fsimpson]
  fsave $f $par(name)-$par(index).fid
}

```

Simpson simulation input file (Oxygen ion with two hydrogen atoms: (1O + 2H)):

```
spinsys {
  channels 17O 1H
  nuclei   17O 1H 1H
  dipole   1 2 113 0 0 0
  dipole   1 3 113 0 180 0
  shift    1 3p 10p 0.5 50 20 10
  quadrupole 1 2 100 0 0 0
}

par {
  variable index    1

  np                200
  spin_rate         16000
  proton_frequency 800e6
  start_operator    I1x
  detect_operator   I1p
  method            direct
  crystal_file      rep320
  gamma_angles     18
  sw                spin_rate/2
  variable tsw      1e6/sw
  verbose           1101
  variable rfF1     50000
  variable rfF2     74600
  variable t180F1   0.5e6/rfF1
  variable t180F2   0.5e6/rfF2
  variable tr1      0.5e6/spin_rate-0.5*t180F1-0.5*t180F2
  variable tr2      0.5e6/spin_rate-t180F2
}

proc pulseseq {} {
  global par
```

```

reset
delay $par(tr2)
pulse $par(t180F2) 0 x $par(rfF2) x
delay $par(tr2)
pulse $par(t180F2) 0 x $par(rfF2) y
store 1

reset
acq
delay $par(tr2)
pulse $par(t180F2) 0 x $par(rfF2) x
delay $par(tr1)
pulse $par(t180F1) $par(rfF1) x 0 x
delay $par(tr1)
pulse $par(t180F2) 0 x $par(rfF2) x
delay $par(tr2)
pulse $par(t180F2) 0 x $par(rfF2) y
store 2
acq

for {set i 2} {$i < $par(np)} {incr i} {
  reset
  prop 1
  prop 2
  prop 1
  store 2
  acq
}
}
proc main {} {
  global par

  set f [fsimpson]
  fsave $f $par(name)-$par(index).fid
}

```

Simpson simulation input file (Oxygen ion with three hydrogen atoms: (1O + 3H)):

```
spinsys {
  channels 17O 1H
  nuclei   17O 1H 1H 1H
  dipole   1 2 113 0 0 0
  dipole   1 3 113 0 120 0
  dipole   1 4 113 0 240 0
  shift    1 3p 10p 0.5 50 20 10
  quadrupole 1 2 100 0 0 0 0
}

par {
  variable index    1

  np                200
  spin_rate         16000
  proton_frequency 800e6
  start_operator    I1x
  detect_operator   I1p
  method            direct
  crystal_file      rep320
  gamma_angles     18
  sw                spin_rate/2
  variable tsw      1e6/sw
  verbose           1101
  variable rfF1     50000
  variable rfF2     74600
  variable t180F1   0.5e6/rfF1
  variable t180F2   0.5e6/rfF2
  variable tr1      0.5e6/spin_rate-0.5*t180F1-0.5*t180F2
  variable tr2      0.5e6/spin_rate-t180F2
}

proc pulseseq {} {
```

```

global par

reset
delay $par(tr2)
pulse $par(t180F2) 0 x $par(rfF2) x
delay $par(tr2)
pulse $par(t180F2) 0 x $par(rfF2) y
store 1

reset
acq
delay $par(tr2)
pulse $par(t180F2) 0 x $par(rfF2) x
delay $par(tr1)
pulse $par(t180F1) $par(rfF1) x 0 x
delay $par(tr1)
pulse $par(t180F2) 0 x $par(rfF2) x
delay $par(tr2)
pulse $par(t180F2) 0 x $par(rfF2) y
store 2
acq

for {set i 2} {$i < $par(np)} {incr i} {
    reset
    prop 1
    prop 2
    prop 1
    store 2
    acq
}
}
proc main {} {
    global par

    set f [fsimpson]

```

fsave \$f \$par(name)-\$par(index).fid

}

1. The ^{17}O enrichment is clearly selective and assumed by the authors in order to reduce enrichment of the oxygen in the bulk only. Moreover, they take advantage of the fact that the targeted surface oxygen sites (at 39 and 42 ppm) present small quadrupolar couplings. On the contrary, they do not observe the $\text{Mg}(\text{OH})_2$ sites. This is bothersome. I understand the difficulty to observe such signal with large second-order quadrupolar couplings. Would ^1H - ^{17}O CP be an option to highlight those sites?

Response: We thank the reviewer for the suggestion. We tried ^1H - ^{17}O CP MAS NMR to explore this species, as suggested by the reviewer (Fig. R3). However, the resonance owing to OH is not observed, which should appear at a low frequency considering that $C_Q = 6.8$ MHz, $\eta = 0$ and $\delta_{\text{iso}} = 25$ ppm for $\text{Mg}(\text{OH})_2$, ref. J. Magn. Reson. 1988, 76, 106.), while only the signal due to oxygen ions in the third coordination shell of H is present (centered at 39 ppm), despite a longer O...H distance. This should be ascribed to the much narrower linewidth and a higher concentration of the species giving rise to the peak at 39 ppm, and a lower concentration and a much wider linewidth for OH species due to the large second-order quadrupolar coupling, as also pointed out by the reviewer.

Fig. R3. $^1\text{H} \rightarrow ^{17}\text{O}$ CP MAS NMR spectrum of NS-773 at 18.8 T. MAS rate: 20 kHz; a recycle delay: 1 s; contact time: 1000 μs . A total of 17200 accumulations were collected.

2. The data and their interpretations represent comprehensive and coherent work. The text is clear with very few (typo) errors and the figures are of high quality. However, if the data are definitely of interest for NMR spectroscopists and to a lesser extent to chemists interested in characterization methods, it seems to me, on the other hand that these results do not fulfill the selection criteria for a publication to Nature Communications, as requested by your editorial board : "...journal aim to represent important advances of significance to specialists within each field". The NMR methods used here are not new, their application to characterize gas adsorption is interesting but lack a broader application (another gas than CO₂ for example).

In my opinion, I would not recommend the publication of this manuscript in Nature Communications; I would rather suggest the submission of a full article (including data from the Supp Info file) in a more specialized journal. I leave that final decision to the editor.

Response: We thank the reviewer for the positive comments on the quality our work. We believe that this work is important and attracts a wide range of readers for the following reasons. First, our ¹⁷O NMR based characterization method can be applied to different oxide nanomaterials. A variety of oxide nanomaterials are important for their multiple applications in a wide range of fields, including adsorption and catalysis, in which gas binding sites play a key role. The ¹⁷O NMR approach demonstrated in this paper, is clearly a *general* technique that can be used to determine the surface structure and gas binding sites for different oxides. Second, different gas adsorption and even catalytic processes can be studied with our approach. In this paper, CO₂ adsorption on MgO nanomaterials is demonstrated and the adsorption of other acidic gas on different basic oxides can be studied in the same manner. It can also be readily extended to investigate basic gas (e.g., NH₃) adsorption on acidic oxides. Adsorption and catalytic processes involve redox reaction can also be monitored. Third, this approach is associated with very high resolution. We know of no other surface spectroscopy with such fine resolution and ability to separate and identify different gas binding sites (i.e., distinguishing sites only different at the third coordination and identifying their

involvement in gas sorption). Furthermore, unlike many other spectroscopic studies of surface, which rely on probe molecules (e.g., trimethylphosphine, or TMP) and may not be suitable for in situ studies, surface structure and gas-surface interaction information are observed directly from oxygen point of view in our approach. We also expect that the method will be combined with signal enhancement techniques, such as dynamic nuclear polarization (DNP), and applied to study oxide materials with larger particle sizes and smaller surface area, further extending the range of applications. Therefore, the results reported represents a significant and methodological advance that should be of interest to workers in the various fields including physical chemistry, surface chemistry, materials chemistry, nanotechnology and NMR spectroscopy, and should attract a broad readership.

Reviewer #2 (Remarks to the Author):

It is good to see a paper with a very straightforward goal that if successful could make a big difference into understanding an important issue. The central premise is that a spectroscopic approach that can elucidate the coordination of molecules on oxide surfaces without the need for a probe molecule or an indirect model could present significant advantages. Here ^{17}O solid-state NMR of ^{17}O surfaces of ^{17}O isotopically-enriched MgO nanosheets is used to investigate the approach. Although this is a model example, the principle looks as though it is well demonstrated. Through different preparations and looking at the effect of introducing CO_2 the different surfaces sites are plausibly identified and the preferential interaction with CO_2 determined. A very clear prediction is made in that the surface OHs present inhibit several of the surrounding surface oxygen sites from coordinating CO_2 .

The use of three magnetic fields for ^{17}O (effectively two for straight spectral comparison - 9.4 and 19.6 T) as well as carrying out REDOR, along with observing ^1H and ^{13}C a high quality NMR methodology is used. There is the back up of other characterisation techniques. There is a good split of the data presented directly in the paper and that given in the Supplementary Information (SI). The only substantive scientific comments are for perhaps some minor expansion of the explanation of some of the technical NMR points given that Nat. Comm. is not a specialist NMR journal some unpacking of the meaning of observation of the relative linewidth at 9.4 and 19.6 T could be added (only a few sentences). Also some of the spectra appear to show some weak spinning sidebands and these could be usefully marked (e.g. Fig. 1 NS-1073 at around 23 ppm?)

Response: We thank the reviewer for these positive comments, helpful suggestions, as well as the time and effort spent in reviewing our paper. We have now marked the peak at 23 ppm in Fig. 1, which is not a spinning side band, and we have added a sentence discussing this peak. We have also explained the differences for linewidths at 9.4 and

19.6 T.

The writing for the most part is very clear so that it is easy to follow the arguments developed. The experiments are clearly described with a judicious use of figures split between the main paper and the SI. A very sensible balance is struck in the referencing which shows a good appreciation. The conclusion one comes to is this is a high quality piece of research that has a good deal of novelty. The issue of being able to elucidate oxide surface coordination sites directly using ^{170}NMR could have a very significant impact. There are several disciplines this approach would be applicable to. Hence on balance I see there are good reasons for publication in Nat. Comm. There are some minor points that need tidying and these are listed below.

Minor corrections

1. p3, line 7, it would be more usual to refer to microscopy than microscopic
2. p3, line 16, approaches is better
3. p3, line 18, should be showed
4. p10, line 1, delete s to give evidence
5. p12, line 1 after the figure caption, should be fractions not factions
6. p16, line 9, probably would be better just to insert ^{170}NMR before method
7. p17, line 5, should be powder

Response: We thank the reviewer for correcting our manuscript. These problems have now been fixed.

Reviewer #3 (Remarks to the Author):

Du et al present an interesting study on the use of ^{17}O enrichment and ^{17}O NMR to understand catalytic behavior in MgO nanosheets. Using ^{17}O NMR and DFT, the authors show that proton poisoning inhibits CO_2 adsorption on nearby surface oxygen sites on MgO. The work has been carefully conducted and provides an advanced understanding of surface chemical reactions without the need for UHV or probe molecules. However, there are several areas that could use additional explanation and analysis, detailed in the comments below:

Response: We thank the reviewer for these positive comments as well as the time and effort spent in reviewing our paper.

1. On page 6, the authors assign the ^{17}O NMR spectra to bulk O in MgO at 47 ppm, and the remaining resonances to surface sites. It would be helpful if the authors could be more quantitative with the data (i.e., what is the % MgO bulk in the two samples shown in Fig 1?). They note that the T1 of the surface vs bulk is very different, based on Figure S4, but do not state the T1 of the bulk? Can the authors comment on this value and whether or not quantitative NMR is possible? They also go on to describe changes in the relative intensity of bulk vs surface between the two samples, but it's not clear if this is due to O_2 diffusion into the bulk at higher annealing temperatures or an actual change in the surface chemistry.

Response: We thank the reviewer for these helpful comments. We analyzed the relative intensities of different peaks in Fig. 1 and the results are shown in Table R1. The extracted percentages of the signals due to oxygen ions in the bulk part of MgO nanosheets are 11 and 7 %, respectively, for NS-773 and NS-1073.

Table R1. The percentages of the bulk (47 ppm) and surface sites (42, 39 or 34 ppm) of NS-773 and NS-1073 in Fig. 1.

Shift/ppm	NS-773	NS-1073
47	11(1)	7(1)
42	20(2)	40(4)
39	65(2)	43(2)
34	3(1)	9(2)

We also fitted the spectral intensities as a function of recycle delay for NS-773, replotted Supplementary Fig. 4, and obtained T_{1s} (Table R2)

Table R2. The longitudinal relaxation time T_1 and stretching exponent β extracted from Supplementary Fig. 4. T_1 is determined by analytical fits using exponential function, $I(t) = I_0(1 - e^{-(t/T_1)^\beta})$, where $I(t)$ and I_0 are the signal intensities at recycle delay t and at equilibrium, respectively.

Shift/ppm	T_1 / s	β
47	2.23	0.50
42+39	1.01	0.55

Quantitative analysis can be performed for the concentrations of different species by simply fitting the spectra. However, due to the different exchange rates between $^{17}\text{O}_2$ and different oxygen species on the surface of MgO nanosheets (at a specific temperature) and the quadrupolar nature of ^{17}O , these results may not be very accurate, especially for quantifying the exact amounts of oxygen ions at different environments. Therefore, we did not include the discussion on the quantitative NMR in the manuscript.

For the two spectra collected for NS-1073, since the enrichment temperatures (623 and 773 K) are lower than the thermal treatment temperature of 1073 K, the surface chemistry should be the same. Therefore, difference in the relative intensity of the signal from the bulk (or the surface) can be attributed to the diffusion of oxygen to the bulk. A higher enrichment temperature leads to more significant diffusion of ^{17}O to the bulk, and thus a stronger signal at 47 ppm. Similar situation is found for the two spectra acquired for NS-773. The explanation has been added following Supplementary Fig. 3.

2. Figure 1: should the resonances at 34 and ~25 ppm be labeled? The peak at 25 ppm is not mentioned in the text.

Response: Thank you for your suggestion. We have now labeled the peaks at 34 and 23 ppm, and we added a sentence discussing these peaks.

3. On page 9, the authors note that the resonance corresponding to adsorbed carbonate is not observed in ^{17}O NMR possibly due to the large linewidth from the second order quadrupolar interaction. If true, it would be interesting to analyze the spectra collected at 18.8 T (e.g. from the REDOR experiments in Fig 3) in more detail or simulate the spectral broadening expected from these sites to discern whether they can be observed.

Response: Thank you for your suggestion. We collected the NMR spectrum at 19.6 T for the sample (NS-1073) adsorbed with CO_2 and there is possibly a very weak and broad resonance at approx. 180 to 90 ppm, which may be ascribed to carbonate species (Fig. R4). The dashed line below shows a possible simulation with $C_Q = 7.5$ MHz, $\eta = 1$ and $\delta_{\text{iso}} = 170$ ppm, which is similar to the quadrupolar parameters extracted from CaCO_3 ($C_Q = 6.97$ MHz, $\eta = 0.98$, ref. Solid State Nucl. Magn. Reson., 1995, 4, 313.), the closest substance with ^{17}O solid-state NMR data in the literature. However, this signal is too weak and we decided to leave it in the response to reviewers, which is still available to the readers.

Fig. R4. ^{17}O single pulse MAS NMR spectra of NS-1073 after $^{13}\text{CO}_2$ chemisorption at 19.6 T. MAS rate: 16 kHz; recycle delay: 5 s. Asterisk shows spinning sideband.

4. Also on page 9, the authors rationalize that CH₃ groups are not present based on ¹H NMR and ¹H-¹³C CPMAS NMR due to the lack of signal. It would be useful to show the full ¹³C NMR spectra in the SI to show the lack of CH₃ groups, as only the carbonate range appears in Fig 2 (similar to how they show an expanded view of the ¹⁷O NMR).

Response: We have added the expanded ¹³C NMR spectra of NS-773 and NS-1073 after ¹³CO₂ chemisorption as Supplementary Fig. 8 (also shown below). Only a single resonance at around 167 ppm in ¹³C NMR spectra due to carbonate species can be observed, while there is no signal owing to CH₃, which should resonate at approx. 50 ppm (ref. Chem. Mater. 1998, 10, 864.).

Supplementary Fig. 8. ¹³C single pulse MAS NMR spectra of NS-773 and NS-1073 after ¹³CO₂ chemisorption.

5. The authors should include error bars for the REDOR measurements reported in Fig 3. The suggested binding sites on the surface are also not entirely consistent with the proposed structures (5.3 Å vs 3.7 Å). Are there other possible surface structures that are not accounted for?

Response: We thank the reviewer for the suggestion. We have replotted Fig. 3(b) in the manuscript with error bars (also shown below). 3.7 Å is the extracted distance between the proton and the oxygen species giving rise to the resonance at 42 ppm according to

the REDOR NMR results, assuming that there is only one proton and one oxygen ion. However, there may be several protons around this oxygen ion (42 ppm) in the real sample, and each proton is coupled to the oxygen ion and contributes to the observed REDOR effects. In this case, even though these different protons are at the 5th coordination shell of a specific oxygen ion, which corresponds to an O-H distance of 5.3 Å, the extracted distance is shorter. In answering the first question raised by Reviewer #1, we did simulations with SIMPSON package considering an oxygen ion coupled with more than one proton and results indicate that it is likely the case.

Fig. 3(b). REDOR fraction ($\Delta S/S_0$) as a function of recoupling time with the schematic surface structure of MgO.

REVIEWER COMMENTS

Reviewer #1 (Remarks to the Author):

See attached pdf

[File follows on the next page]

Reviewer #3 (Remarks to the Author):

My concerns have been addressed by the authors and the manuscript is now suitable for publication.

Response to Reviewers

Reviewer #1 (Remarks to the Author):

Du and coauthors present a structural study of oxide nanomaterials using advanced methods in solid-state NMR spectroscopy in order to identify at the surface of MgO nanomaterials, those sites that are primarily involved into the adsorption of gas. Here they use CO₂ adsorption as a model reaction for further extrapolation of their NMR protocol to other gas adsorption (not investigated in the present paper). The authors focus on the use of ¹⁷O NMR as a sensitive probe, a nucleus that has gained large interest through the last two decades especially with very high magnetic fields now readily available. The present study is first based on the identification and assignment of two resonances at 39 and 42 ppm attributed to surface oxygen sites. Further, the authors use a more advanced NMR technique, the ¹H-¹⁷O REDOR NMR pulse sequence, to determine proton-oxygen distance and thus, propose a model for the surface of their MgO nanomaterials where the resonance at 39 ppm would be associated

to hydroxyl oxygen. Previously, by investigating two MgO materials, calcined at two temperatures (773 and 1073 K), Du et al. showed that the oxygen site at 42 ppm is the one most affected by the CO₂ adsorption.

I have a few questions about the methods and protocol.

- The authors consider a model with an isolated hydroxyl group in order to interpret their REDOR curves. Would it be more realistic to consider a multispin system (rather than a spin pair) to explain the REDOR curves especially at long recoupling time (> 1.5 ms)?

Response: We thank the reviewer for the helpful suggestion. Of course, a simulation considering more than two spins should be better. As requested, we tried to use SIMPSON software (M. Bak, J. T. Rasmussen, N. C. Nielsen, SIMPSON: A General Simulation Program for Solid-State NMR Spectroscopy, J. Magn. Reson., 2000, 147, 296) to simulate the REDOR curves for one oxygen atom with one or more hydrogen

atoms around. As can be imagined, simulations considering such multi-spin system can be quite challenging and the simulation is expected to take much longer time with each additional spin, which was also explained in the literature (J. Magn. Reson. 2006, 178, 248.). We managed to do simulations with up to 3 hydrogen atoms in the fifth coordination shell of one oxygen atom (Fig. R1 and R2, and the SIMPSON simulation input files are shown below. The simulation considering 3 hydrogen atoms already took several days to finish and the simulation considering 4 hydrogen atoms failed). It is clear that with more hydrogen atoms around, the REDOR fraction increases steadily, indicating that the effects from each additional hydrogen atom add up (spheres in Fig. R2). A clear trend can be seen that a higher REDOR fraction can be expected with 4 or more hydrogen atoms. From trend of the obtained simulation results, it is expected that one oxygen atom with around 6 hydrogen atoms at the fifth coordination shell should generate a REDOR curve similar to the experimental data for the peak at 42 ppm in NS-773. Because the simulations considering more than 3 hydrogen atoms failed, we decided to leave the related results in the response to reviewers, which is still available to the readers.

I thank the authors for the effort in running Simpson calculations. The results are those expected. In particular, it shows the possible influence of hydrogen atoms in the fifth coordination shell for recoupling times larger than 1.5-2 ms. Since the distance can be roughly evaluated with the first part of the REDOR curve (< 2 ms), the estimation given by the authors can probably be validated. The precision of the distance measurement could also be provided ?

But, the present data also prove the complexity of such approach. The system here is quite simple with a defined spin pair and the REDOR approach is mostly dedicated to systems presenting a spin pair. One of the reasons is due to the SIMPSON calculation time, as mentioned by the authors in the present response. The approach has its limits when the surface has multiple sources of hydrogen. I wanted to lower the expectations of using REDOR, even if I understand that this is not the central piece of this work.

1. The ^{17}O enrichment is clearly selective and assumed by the authors in order to reduce enrichment of the oxygen in the bulk only. Moreover, they take advantage of the fact that the targeted surface oxygen sites (at 39 and 42 ppm) present small quadrupolar couplings. On the contrary, they do not observe the $\text{Mg}(\text{OH})_2$ sites.

This is bothersome. I understand the difficulty to observe such signal with large second-order quadrupolar couplings. Would ^1H - ^{17}O CP be an option to highlight those sites?

Response: We thank the reviewer for the suggestion. We tried ^1H - ^{17}O CP MAS NMR to explore this species, as suggested by the reviewer (Fig. R3). However, the resonance owing to OH is not observed, which should appear at a low frequency considering that $C_Q = 6.8$ MHz, $\eta = 0$ and $\delta_{\text{iso}} = 25$ ppm for $\text{Mg}(\text{OH})_2$, ref. J. Magn. Reson. 1988, 76, 106.), while only the signal due to oxygen ions in the third coordination shell of H is present (centered at 39 ppm), despite a longer O...H distance. This should be ascribed to the much narrower linewidth and a higher concentration of the species giving rise to the peak at 39 ppm, and a lower concentration and a much wider linewidth for OH species due to the large second-order quadrupolar coupling, as also pointed out by the reviewer.

Fig. R3. ^1H → ^{17}O CP MAS NMR spectrum of NS-773 at 18.8 T. MAS rate: 20 kHz; a recycle delay: 1 s; contact time: 1000 μs . A total of 17200 accumulations were collected.

I thank the authors for providing the additional CP MAS NMR spectrum. However, I would expect more investigations based on this spectrum.

- Actually, CP involving a quadrupolar nucleus, moreover with (relatively) large quadrupolar coupling, requires different Hartmann-Hahn (HH) conditions than those for sites at 39 and 42 ppm (small quadrupolar couplings). This is due to the selective aspect of CP to quadrupolar nucleus (and to quadrupolar coupling constants).

- o What HH conditions were used for the CP experiment ?

- Authors should also try other experiments such as INEPT, HMQC for example, to avoid cross-polarization of a quadrupolar nucleus. There are many articles showing the advantage of such approach wrt CP. They show much less selectivity to quadrupolar couplings.
- Why ^1H - ^{17}O CP MAS NMR spectrum only shows a signal at 39 ppm, not the one at 42 ppm (distance O-H of approximately 3.0 Å and 3.7 Å, respectively). Both sites show relatively small quadrupolar couplings (as mentioned in the text by comparing ^{17}O linewidth at 9.4 and 19.6 T). I would expect both signals to appear on the spectrum.

I am questioning the robustness of the methodology presented here. If the authors want to have readers of Nature Communications to perform the same kind of approach to other (more complex) systems and reactivity, the experiments should be robust. For my part, I believe that it fails.

2. The data and their interpretations represent comprehensive and coherent work. The text is clear with very few (typo) errors and the figures are of high quality. However, if the data are definitely of interest for NMR spectroscopists and to a lesser extent to chemists interested in characterization methods, it seems to me, on the other hand that these results do not fulfill the selection criteria for a publication to Nature Communications, as requested by your editorial board : "...journal aim to represent important advances of significance to specialists within each field". The NMR methods used here are not new, their application to characterize gas adsorption is interesting but lack a broader application (another gas than CO_2 for example).

In my opinion, I would not recommend the publication of this manuscript in Nature Communications; I would rather suggest the submission of a full article (including data from the Supp Info file) in a more specialized journal. I leave that final decision to the editor.

Response: We thank the reviewer for the positive comments on the quality of our work. We believe that this work is important and attracts a wide range of readers for the

following reasons. First, our ^{17}O NMR based characterization method can be applied to different oxide nanomaterials. A variety of oxide nanomaterials are important for their multiple applications in a wide range of fields, including adsorption and catalysis, in which gas binding sites play a key role. The ^{17}O NMR approach demonstrated in this paper, is clearly a *general* technique that can be used to determine the surface structure and gas binding sites for different oxides.

Here, it is demonstrated that a ^{17}O NMR signal can be detected for sites experiencing small quadrupolar couplings. Maybe other oxide nanomaterials will present sites with large couplings. It is then possible that some sites (with large quadrupolar coupling constants, like Mg-O-H here) may not be detected. Therefore, I do not consider the approach as a general technique.

Second, different gas adsorption and even catalytic processes can be studied with our approach. In this paper, CO_2 adsorption on MgO nanomaterials is demonstrated and the adsorption of other acidic gas on different basic oxides can be studied in the same manner. It can also be readily extended to investigate basic gas (e.g., NH_3) adsorption on acidic oxides. Adsorption and catalytic processes involve redox reaction can also be monitored. Third, this approach is associated with very high resolution. We know of no other surface spectroscopy with such fine resolution and ability to separate and identify different gas binding sites (i.e., distinguishing sites only different at the third coordination and identifying their involvement in gas sorption). Furthermore, unlike many other spectroscopic studies of surface, which rely on probe molecules (e.g., trimethylphosphine, or TMP) and may not be suitable for in situ studies, surface structure and gas-surface interaction information are observed directly from oxygen point of view in our approach.

For Nature Communications, I would expect other catalytic processes to be investigated, not just CO_2 adsorption on MgO nanomaterials. Maybe the authors should extend the application of such approach and submit their article with additional data.

We also expect that the method will be combined with signal enhancement techniques, such as dynamic nuclear polarization (DNP), and applied to study oxide materials with larger particle sizes and smaller surface area, further extending the range of applications.

Yes I agree that DNP may improve ^{17}O NMR sensitivity. I do not put doubts on the significant progress brought by this technique. But the authors are aware that DNP requires chemical manipulation (and probably modification) of the surface of the materials. We should not give vain hope to a broad readership with no precise knowledge on the actual limits of DNP.

Therefore, the results reported represents a significant and methodological advance that should be of interest to workers in the various fields including physical chemistry, surface chemistry, materials chemistry, nanotechnology and NMR spectroscopy, and should attract a broad readership.

The additional data and information brought by the authors have not changed my mind. I think that the present article is interesting at this stage for readers of more specialized journals. The authors should extend the study to prove that the methodology is indeed a real breakthrough.

Response to Reviewer 1

Reviewer #1 (Remarks to the Author):

Du and coauthors present a structural study of oxide nanomaterials using advanced methods in solid-state NMR spectroscopy in order to identify at the surface of MgO nanomaterials, those sites that are primarily involved into the adsorption of gas. Here they use CO₂ adsorption as a model reaction for further extrapolation of their NMR protocol to other gas adsorption (not investigated in the present paper). The authors focus on the use of ¹⁷O NMR as a sensitive probe, a nucleus that has gained large interest through the last two decades especially with very high magnetic fields now readily available. The present study is first based on the identification and assignment of two resonances at 39 and 42 ppm attributed to surface oxygen sites. Further, the authors use a more advanced NMR technique, the ¹H-¹⁷O REDOR NMR pulse sequence, to determine proton-oxygen distance and thus, propose a model for the surface of their MgO nanomaterials where the resonance at 39 ppm would be associated

to hydroxyl oxygen. Previously, by investigating two MgO materials, calcined at two temperatures (773 and 1073 K), Du et al. showed that the oxygen site at 42 ppm is the one most affected by the CO₂ adsorption.

I have a few questions about the methods and protocol.

- The authors consider a model with an isolated hydroxyl group in order to interpret their REDOR curves. Would it be more realistic to consider a multispin system (rather than a spin pair) to explain the REDOR curves especially at long recoupling time (> 1.5 ms)?

Response: We thank the reviewer for the helpful suggestion. Of course, a simulation considering more than two spins should be better. As requested, we tried to use SIMPSON software (M. Bak, J. T. Rasmussen, N. C. Nielsen, SIMPSON: A General Simulation Program for Solid-State NMR Spectroscopy, *J. Magn. Reson.*, 2000, 147, 296) to simulate the REDOR curves for one oxygen atom with one or more hydrogen atoms around. As can be imagined, simulations considering such multi-spin system can be quite challenging and the simulation is expected to take much longer time with each additional spin, which was also explained in the literature (*J. Magn. Reson.* 2006, 178, 248.). We managed to do simulations with up to 3 hydrogen atoms in the fifth coordination shell of one oxygen atom (Fig. R1 and R2, and the SIMPSON simulation input files are shown below. The simulation considering 3 hydrogen atoms already took several days to finish and the simulation considering 4 hydrogen atoms failed). It is clear that with more hydrogen atoms around, the REDOR fraction increases steadily, indicating that the effects from each additional hydrogen atom add up (spheres in Fig.

R2). A clear trend can be seen that a higher REDOR fraction can be expected with 4 or more hydrogen atoms. From trend of the obtained simulation results, it is expected that one oxygen atom with around 6 hydrogen atoms at the fifth coordination shell should generate a REDOR curve similar to the experimental data for the peak at 42 ppm in NS-773. Because the simulations considering more than 3 hydrogen atoms failed, we decided to leave the related results in the response to reviewers, which is still available to the readers.

I thank the authors for the effort in running Simpson calculations. The results are those expected. In particular, it shows the possible influence of hydrogen atoms in the fifth coordination shell for recoupling times larger than 1.5-2 ms. Since the distance can be roughly evaluated with the first part of the REDOR curve (< 2 ms), the estimation given by the authors can probably be validated. The precision of the distance measurement could also be provided ?

But, the present data also prove the complexity of such approach. The system here is quite simple with a defined spin pair and the REDOR approach is mostly dedicated to systems presenting a spin pair. One of the reasons is due to the SIMPSON calculation time, as mentioned by the authors in the present response. The approach has its limits when the surface has multiple sources of hydrogen. I wanted to lower the expectations of using REDOR, even if I understand that this is not the central piece of this work.

Response: We thank the reviewer for the helpful suggestion. If there is no motion involved, the precision is mostly dependent on the signal/noise ratio of the spectrum, which is determined by factors such as external magnetic field, number of data acquisition and (¹⁷O) enrichment level. Based on the signal/noise ratio of our experimental data, we estimate that the precision of the O-H distance measured is ± 0.3 Å, and we have added the information in the SI (Supplementary Fig. 10.). Of course, the precision of the distance measured with REDOR can be much better (<0.1 Å, e.g., R. C. Anderson, J. Am. Chem. Soc., 1995, 117, 10546)

The reviewer also raised a very important point on the application of REDOR. REDOR is very convenient when only a spin pair is considered. With two spin pairs or more, REDOR combined with simulations using programs such as SIMPSON is required, while it can be time-consuming and challenging when there are a lot of spin pairs. However, it is very difficult to extract detailed structure information (e.g., bonding information and internuclear distances up to 3rd to 5th coordination) on the surface of oxide, and REDOR (or any other double resonance NMR technique) is still one of the very few methods possible to deal with such difficult problems. Therefore, we consider that REDOR can provide important and helpful information for the surface of oxides, and it is worth a try.

1. The ^{17}O enrichment is clearly selective and assumed by the authors in order to reduce enrichment of the oxygen in the bulk only. Moreover, they take advantage of the fact that the targeted surface oxygen sites (at 39 and 42 ppm) present small quadrupolar couplings. On the contrary, they do not observe the $\text{Mg}(\text{OH})_2$ sites.

This is bothersome. I understand the difficulty to observe such signal with large second-order quadrupolar couplings. Would ^1H - ^{17}O CP be an option to highlight those sites?

Response: We thank the reviewer for the suggestion. We tried $^1\text{H} \rightarrow ^{17}\text{O}$ CP MAS NMR to explore this species, as suggested by the reviewer (Fig. R3). However, the resonance owing to OH is not observed, which should appear at a low frequency considering that $C_Q = 6.8$ MHz, $\eta = 0$ and $\delta_{\text{iso}} = 25$ ppm for $\text{Mg}(\text{OH})_2$, ref. J. Magn. Reson. 1988, 76, 106.), while only the signal due to oxygen ions in the third coordination shell of H is present (centered at 39 ppm), despite a longer O...H distance. This should be ascribed to the much narrower linewidth and a higher concentration of the species giving rise to the peak at 39 ppm, and a lower concentration and a much wider linewidth for OH species due to the large second-order quadrupolar coupling, as also pointed out by the reviewer.

Fig. R3 $^1\text{H} \rightarrow ^{17}\text{O}$ CP MAS NMR spectrum of NS-773 at 18.8 T. MAS rate: 20 kHz; recycle delay: 1 s; contact time: 1000 μs . A total of 17200 accumulations were collected.

I thank the authors for providing the additional CP MAS NMR spectrum. However, I would expect more investigations based on this spectrum.

- Actually, CP involving a quadrupolar nucleus, moreover with (relatively) large quadrupolar coupling, requires different Hartmann-Hahn (HH) conditions than those for sites at 39 and 42 ppm (small quadrupolar couplings). This is due to the selective aspect of CP to quadrupolar nucleus (and to quadrupolar coupling constants).
- o What HH conditions were used for the CP experiment ?
- o Authors should also try other experiments such as INEPT, HMQC

for example, to avoid cross-polarization of a quadrupolar nucleus. There are many articles showing the advantage of such approach wrt CP. They show much less selectivity to quadrupolar couplings.

Response: We understand the reviewer's concern. The cross polarization spin lock pulses were optimized on Mg(OH)₂, which has a large C_Q and is the most relevant sample compared to ours. The ¹⁷O spin lock pulse has an RF power of approx. 47 kHz (measured for the central transition nutation rate), while ¹H spin lock pulse has a linear RF power ramp from 45 to 90 kHz (i.e., averaged at 67.5 kHz). Therefore, considering that the spinning rate is 20 kHz, this Hartmann-Hahn (HH) condition was set for the signal with a large quadrupolar coupling (i.e., the possible signal due to oxygen species directly bound to hydrogen) rather than the resonance with small quadrupolar interactions (i.e., 39 and 42 ppm, corresponding oxygen atoms in the 3rd coordination shell of hydrogen and further away, respectively).

As requested by the reviewer, we also performed ¹H→¹⁷O INEPT NMR experiment on the sample, which explores through bond connectivity (Fig. RR-1). However, no signal is observed, indicating the oxygen species directly bound to hydrogen has a very low concentration. For the species giving rise to the peaks at 39 and 42 ppm, hydrogen atoms are at the 3rd coordination shell or further away, which should have much smaller J coupling constants. We did calculations and confirmed that the J coupling constant between oxygen ion and hydrogen atoms at its 3rd coordination shell is as small as 1 Hz (Fig. RR-2). Therefore, these peaks are not expected to appear in the INEPT NMR spectrum, which explains that no peak is observed in the ¹H→¹⁷O INEPT NMR spectrum. Again, we believe that the absence of the signal owing to oxygen directly connected to hydrogen in CP MAS NMR spectrum is mainly because of the low concentration, while the large quadrupolar coupling interaction may also play a role.

Fig. RR-1 ¹H→¹⁷O INEPT NMR of NS-773 at 9.4 T. Solid π (2.2 μ s) and $\pi/2$ (1.1 μ s) pulses were used for ¹⁷O and the evolution time (τ , the time between π and $\pi/2$ pulses) = 2.94 ms. MAS rate: 16 kHz; recycle delay: 2 s. A total of 14400 accumulations were collected.

Fig. RR-2 Calculations of the values of J coupling constants on the surface of MgO (111). (a) Side view and (b) top view with the values of J coupling constants between the adsorbed H atom (white) and specific O species (red). All the calculations are accomplished with the Gaussian 03 package. The values of J coupling constants were obtained using the gauge-including atomic orbital (GIAO) method (R. Ditchfield, *Mol. Phys.*, 1974, 27, 789). All the optimizations were performed using the hybrid *ab initio* density-functional B3LYP method (A. D. Becke, *J. Chem. Phys.*, 1992, 97, 9173; C. Lee et al., *Phys. Rev. B*, 1988, 37, 785). We optimized the geometry of the cluster (MgO)₃₃ using the 6-31G basis set (J. S. Binkley et al., *J. Am. Chem. Soc.*, 1980, 102, 939). In order to achieve a higher accuracy, the PCJ-1 basis set was used in calculating the values of J coupling constants (F. Jensen, *Theoret. Chem. Acc.*, 2010, 126, 371). The J coupling constant calculated between O and its directly bound H is approx. 100 Hz, while the values of J coupling constants between H and O atoms at the 3rd coordination shell of H are as small as approx. 1 Hz. This value further decreases to 0.1 Hz if the H is at the 5th coordination shell.

- Why ^1H - ^{17}O CP MAS NMR spectrum only shows a signal at 39 ppm, not the one at 42 ppm (distance O-H of approximately 3.0 Å and 3.7 Å, respectively). Both sites show relatively small quadrupolar couplings (as mentioned in the text by comparing ^{17}O linewidth at 9.4 and 19.6 T). I would expect both signals to appear on the spectrum.

I am questioning the robustness of the methodology presented here. If the authors want to have readers of Nature Communications to perform the same kind of approach to other (more complex) systems and reactivity, the experiments should be robust. For my part, I believe that it fails.

Response: We thank the reviewer for raising another important factor of this approach, robustness. Fig. RR-3, which is the enlarged view of the spectrum shown in Fig. R-3, exhibits a major peak at 39 with a shoulder at 42 ppm. The peak at 39 ppm is stronger than the peak at 42 ppm, due to both a closer O...H distance and a higher concentration of the corresponding species. Therefore, both peaks at 39 and 42 do appear in the spectrum, and these results actually prove the robustness of the methodology.

Fig. RR-3 Enlarged view of Fig. R3.

2. The data and their interpretations represent comprehensive and coherent work. The text is clear with very few (typo) errors and the figures are of high quality. However, if the data are definitely of interest for NMR spectroscopists and to a lesser extent to chemists interested in characterization methods, it seems to me, on the other hand that these results do not fulfill the selection criteria for a publication to Nature Communications, as requested by your editorial board : "...journal aim to represent important advances of significance to specialists within each field". The NMR methods used here are not new, their application to characterize gas adsorption is interesting but lack a broader application (another gas than CO_2 for example).

In my opinion, I would not recommend the publication of this manuscript in Nature Communications; I would rather suggest the submission of a full article (including data from the Supp Info file) in a more specialized journal. I leave that final decision to the editor.

Response: We thank the reviewer for the positive comments on the quality our work. We believe that this work is important and attracts a wide range of readers for the following reasons. First, our ^{17}O NMR based characterization method can be applied to different oxide nanomaterials. A variety of oxide nanomaterials are important for their multiple applications in a wide range of fields, including adsorption and catalysis, in which gas binding sites play a key role. The ^{17}O NMR approach demonstrated in this paper, is clearly a general technique that can be used to determine the surface structure and gas binding sites for different oxides.

Here, it is demonstrated that a ^{17}O NMR signal can be detected for sites experiencing small quadrupolar couplings. Maybe other oxide nanomaterials will present sites with large couplings. It is then possible that some sites (with large quadrupolar coupling constants, like Mg-O-H here) may not be detected. Therefore, I do not consider the approach as a general technique.

Response: The reviewer's concern is the broad applicability of the method reported, which is of course very important. The "oxide nanomaterials" concept in the title of our manuscript usually refer to "metal oxide nanomaterials". Most of the simple metal oxides do have a relatively small C_Q . The values of C_{QS} of ^{17}O in binary oxides are usually about 1 MHz or less, leading to sharp ^{17}O NMR peaks, for example, BeO, MgO, CaO, SrO, BaO, ZnO, TiO₂, PbO, ZrO₂, HfO₂, La₂O₃, CeO₂, Cu₂O and Ag₂O (G. L. Turner, S. E. Chung, E. Oldfield, J. Mag. Reson., 1985, 64, 316.; T. J. Bastow, S. N. Stuart, Chem. Phys. 1990, 143, 459). A few metal oxides have medium sized C_{QS} of about 1 to 3 MHz, resulting in wider resonances, including Al₂O₃, Ti₂O₃ and SnO (T. J. Bastow, S. N. Stuart, Chem. Phys. 1990, 143, 459.), and it is still quite easy to detect these signals at medium external fields (e.g., 9.4 T or 400 MHz for ^1H). Many ternary oxides also have small C_{QS} (~1 MHz or less), such as CaTiO₃, SrTiO₃, Sr₂TiO₄, Na₂ZrO₃, CaZrO₃, Sr₂SnO₄, KMnO₄ and K₂WO₄ (T. J., Bastow, P. J. Dirken, M. E. Smith, H. J. Whitfield, J. Phys. Chem., 1996, 100, 18539; S. Schramm, E. Oldfield, J. Am. Chem. Soc., 1984, 106, 2502). The small to medium C_{QS} can be ascribed to the strong ionic character of the metal oxides. A relationship was already established in 1980s between the size of C_Q and the ionic character (I , related to the electronegativity of the element M) of the M-O bond, which can be expressed as: C_Q (MHz) = $-0.203I(\%) + 14.78$ (see the "Figure 3" below from Schramm and Oldfield, J. Am. Chem. Soc., 1984, 106, 2502). The smaller the electronegativity, the smaller the C_Q , and most of the metal oxides do fall in the region with small to medium C_{QS} . Therefore, oxygen ions in

most of the diamagnetic metal oxides can be easily observed with ^{17}O NMR. Those oxygen species with a large quadrupolar coupling interaction are oxygen ions connected to elements with a larger electronegativity (N, C, H etc), and these materials are not what “oxide nanomaterials” usually refer to. To make this clear, we added “metal” before “oxide nanomaterials” in the title and the main text.

Figure 3. Plot of ^{17}O electric quadrupole coupling constant (e^2qQ/h , MHz) vs. average percent ionic character for a series of A–O–B fragments, where A and B are cations. The experimental e^2qQ/h values are for a series of oxides or simple oxyanions. The percent ionic character is the arithmetic mean of the single-bond values obtained from the Pauling electronegativities of elements A, B, and O. Species more covalent than SiO_2 are best studied by NQR or wide-line NMR methods, while more ionic systems are amenable to “high-resolution” solid-state MASS and VASS NMR techniques. The straight line is the least-squares fit of the data to eq 1. Compounds used were as follows: (a) *N*-methylsyndone, (b) tetrahydropyran, (c) xanthene, (d) tetrachloro-hydroquinone, (e) 2,5-dichloroquinone, *N*-methylsyndone, *p*-chlorophenol, (g) normal hexagonal ice, (h) B_2O_3 , (i) low cristobalite, (j) diopside, (k) forsterite, (l) Al_2O_3 , (m) zinc oxide, (n) potassium tungstate, (o) magnesium oxide.

(From Schramm and Oldfield, *J. Am. Chem. Soc.*, 1984, 106, 2502)

The O-H species the reviewer mentioned do often have a large C_Q . However, they are usually not the major species on the surface oxide materials and the main reason these sites are not present in our spectrum is the low concentration. We believe that a higher concentration of these species will also be conveniently observable with our approach, e.g., hydroxide (Fig. RR-4, ^{17}O NMR of $\text{Mg}(\text{OH})_2$). However, with a high OH concentration on the surface, we would call the material hydroxide rather than an oxide. Thus, the question the reviewer raised should not be a big problem for oxide materials.

Fig. RR-4 Single pulse ^{17}O MAS NMR spectrum of ^{17}O enriched $\text{Mg}(\text{OH})_2$ at 9.4 T. MAS rate: 15 kHz; recycle delay: 10 s. A total of 64 accumulations were collected.

Although a small fraction of metal oxides may have large C_{QS} and it would be difficult to study these systems at medium external fields, as discussed above, our approach is still suitable for a variety of different metal oxides. In addition, the problem may also be possibly solved at higher external fields. Therefore, we still consider our approach a *general* approach.

Second, different gas adsorption and even catalytic processes can be studied with our approach. In this paper, CO_2 adsorption on MgO nanomaterials is demonstrated and the adsorption of other acidic gas on different basic oxides can be studied in the same manner. It can also be readily extended to investigate basic gas (e.g., NH_3) adsorption on acidic oxides. Adsorption and catalytic processes involve redox reaction can also be monitored. Third, this approach is associated with very high resolution. We know of no other surface spectroscopy with such fine resolution and ability to separate and identify different gas binding sites (i.e., distinguishing sites only different at the third coordination and identifying their involvement in gas sorption). Furthermore, unlike many other spectroscopic studies of surface, which rely on probe molecules (e.g., trimethylphosphine, or TMP) and may not be suitable for in situ studies, surface structure and gas-surface interaction information are observed directly from oxygen point of view in our approach.

For Nature Communications, I would expect other catalytic processes to be investigated, not just CO_2 adsorption on MgO nanomaterials. Maybe the authors should extend the application of such approach and submit their article with additional data.

Response: This manuscript is entitled “Identification of Gas Adsorption Sites on Oxide Nanomaterials with Solid-state NMR Spectroscopy”, which is focused on the study of gas adsorption sites on metal oxides, and the approach presented is directly relevant to adsorption and catalysis applications. Differentiating sites on the surface of materials and determining gas adsorption sites is non-trivial. We show that our approach is able to distinguish surface sites of MgO only different in their 3rd coordination and prove these sites show distinct adsorption properties. To the best of our knowledge, we don’t know any other surface spectroscopy that can provide such fine resolution and chemical information. It means surface chemistry of oxides can be performed at a new level with our approach. Therefore, we believe these results merit publication in Nature Communications.

We agree with the reviewer that demonstrating this method can be used to study catalytic process on oxides would be very useful. However, a lot of data and information need to be added and it is very hard to combine those into a single paper. Furthermore, we have tested our MgO nanosheets on a couple of catalytic reactions including oxidative coupling of methane (OCM), however, these materials do not show very good catalytic performances. Therefore, we expect much longer time than 6 months to fully explore the catalytic performances of our MgO nanosheet samples and perform related

NMR experiments, or do a full and careful investigation on the surface structure and catalytic processes over another oxide catalyst with our NMR approach. We do hope that the reviewer can agree that investigations of catalytic processes is beyond the scope of this paper.

We also expect that the method will be combined with signal enhancement techniques, such as dynamic nuclear polarization (DNP), and applied to study oxide materials with larger particle sizes and smaller surface area, further extending the range of applications.

Yes I agree that DNP may improve ^{17}O NMR sensitivity. I do not put doubts on the significant progress brought by this technique. But the authors are aware that DNP requires chemical manipulation (and probably modification) of the surface of the materials. We should not give vain hope to a broad readership with no precise knowledge on the actual limits of DNP.

Response: We are aware that DNP requires adding polarizing agents which potentially change the surface of the materials. However, in addition to signal enhancement (up to 2 orders of magnitude), DNP has already been developed as a powerful surface characterization tool in the past a few years, i.e., Dynamic Nuclear Polarization Surface Enhanced NMR Spectroscopy (DNP-SENS). (L. Emsley*, *Acc. Chem. Res.*, 2013, 46, 1942.) This field has attracted a lot of attention and the surface of many materials have been studied with DNP-SENS, including heterogenous catalysts (e. g., *ACS Catal.* 2015, 5, 7055; *Angew. Chem., Int. Ed.* 2016, 55, 4743; *Chem. Sci.* 2017, 8, 416; *Nat. Commun.* 2019, 10, 5420; *J. Am. Chem. Soc.* 2020, 142, 18936). Therefore, it is feasible to selectively study the surface of solids with DNP. This was also supported by a very recent review paper published in *Nature Reviews Methods Primers* which states “The introduction of DNP surface-enhanced NMR spectroscopy (DNP SENS) has largely solved this problem (low sensitivity associated with ^{17}O NMR studies of surface of oxides) in the past decade.” (*Nat. Rev. Methods Primers*, 2021, 1(1), 2. <https://doi.org/10.1038/s43586-020-00002-1>, we have also added this new reference in the revised main text). Thus, at least there is a good possibility that DNP can be combined to study the surface of oxides.

Nonetheless, we understand the reviewer’s concerns and have rephrased the related discussion. Now it reads “This ^{17}O NMR method may also be combined with signal enhancement techniques, such as dynamic nuclear polarization⁴⁶⁻⁵⁰, to study materials with larger sizes and smaller surface area, and thus be extended to a variety of other metal oxides to investigate gas adsorption processes, and help design related materials with improved properties for adsorption and catalysis.”

Therefore, the results reported represents a significant and methodological advance that should be of interest to workers in the various fields including physical chemistry, surface chemistry, materials chemistry, nanotechnology and NMR spectroscopy, and

should attract a broad readership.

The additional data and information brought by the authors have not changed my mind. I think that the present article is interesting at this stage for readers of more specialized journals. The authors should extend the study to prove that the methodology is indeed a real breakthrough.

Response: This work demonstrates a sensitive characterization method that can distinguish sites on the surface of an oxide, different only in the third coordination shell, and can show that the gas molecules are adsorbed on only one type of sites. We know of no other surface spectroscopy with such fine resolution and ability to separate between different sites and their adsorption abilities (c.f., XPS and O-edge NEXAFS). In addition, this method is based on NMR, meaning that the information obtained is representative of the whole sample (unlike various electron microscopy techniques). We also show the approach is robust, and should be applied to a variety of metal oxide nanomaterials.

Furthermore, in a very recent study, we found that the metal binding sites on metal/oxide materials, which are widely used as heterogeneous catalysts, can also be explored with this approach. The ^{17}O NMR spectrum of Au/MgO is compared to the parent MgO nanosheets (NS-1073) in Supplementary Fig. 17. The spectrum of Au/MgO exhibits a broad resonance with contributions from both the peaks at 42 and 39 ppm, along with a sharp component at 47 ppm. The latter corresponds to the oxygen ions in the bulk part of the material, which has almost the same intensity compared to the parent MgO (NS-1073). It is clear that the peak at 42 ppm is associated with a more significant decrease (more than 90%) in intensity than the resonance at 39 ppm (less than around 25%) after Au loading. The decrease in spectral intensity may be ascribed to change of chemical environment due to the presence of Au. Therefore, this preliminary result implies that bare $\text{O}_{3\text{C}}$ sites on the surface without proton in the 3rd coordination shell may more likely to be the metal binding sites, as compared to the $\text{O}_{3\text{C}}$ sites with nearby protons. We have added this result in the Supplementary Information with related discussions.

Supplementary Fig. 17. ^{17}O single pulse MAS NMR data of MgO (NS-1073) and corresponding Au/MgO obtained at 9.4 T. MAS rate: 16 kHz; recycle delay: 5 s.

Therefore, the approach presented in this paper has broad applicability, which is indeed a significant and methodological advance, and a real breakthrough.

Editorial Note: Reviewer #1 was unavailable to review this round, so Reviewer #3 was asked to comment on the previous concerns.

REVIEWER COMMENTS

Reviewer #3 (Remarks to the Author):

The authors have addressed most of the reviewers' concerns regarding the limitations of REDOR and clarified aspects of the CP measurements.

From these additional experiments, the authors affirm that this approach is mostly limited to O surface sites with small C_q values. The authors argue that most of the surface terminal sites on metal oxides are indeed low C_q (but this is based on literature values for bulk materials), and I'm not sure that is true for the surface sites with lower coordination number and dangling bonds. Many oxides are believed to be terminated with O-H groups with larger C_q values that may not be detected. Of course, the surface chemistries are critical for catalytic performance.

The reason for a lack of detection is that these species, even in higher abundance, exhibit line broadening due to the large C_q values, as demonstrated in Fig RR-4 (i.e., small amounts of signal are spread over a larger ppm range compared to small C_q sites).

I leave the decision to publish this up to the editor. Both the reviewer and the authors have valid points (reviewer says this is not general and can only be used for small C_q sites, authors argue many surface sites do have small C_q (based on this study and prior literature for bulk materials)). I have to say I agree a bit more with the reviewer in that the lack of O-H is difficult to validate experimentally, especially if O-H cannot be readily detected. In other words, it is hard to argue that something is/is not there in light of these challenges. This argument could be strengthened by complementing the NMR measurements with other experimental or computational methods.

I don't think this negates the fact that this study is interesting and useful, but that the claims must be tempered in light of this information (i.e., the fact that O-H cannot be readily detected needs to be disclosed).

Response to Reviewer

Reviewer #3

The authors have addressed most of the reviewers' concerns regarding the limitations of REDOR and clarified aspects of the CP measurements.

From these additional experiments, the authors affirm that this approach is mostly limited to O surface sites with small C_q values. The authors argue that most of the surface terminal sites on metal oxides are indeed low C_q (but this is based on literature values for bulk materials), and I'm not sure that is true for the surface sites with lower coordination number and dangling bonds. Many oxides are believed to be terminated with O-H groups with larger C_q values that may not be detected. Of course, the surface chemistries are critical for catalytic performance.

The reason for a lack of detection is that these species, even in higher abundance, exhibit line broadening due to the large C_q values, as demonstrated in Fig RR-4 (i.e., small amounts of signal are spread over a larger ppm range compared to small C_q sites).

I leave the decision to publish this up to the editor. Both the reviewer and the authors have valid points (reviewer says this is not general and can only be used for small C_q sites, authors argue many surface sites do have small C_q (based on this study and prior literature for bulk materials)). I have to say I agree a bit more with the reviewer in that the lack of O-H is difficult to validate experimentally, especially if O-H cannot be readily detected. In other words, it is hard to argue that something is/is not there in light of these challenges. This argument could be strengthened by complementing the NMR measurements with other experimental or computational methods.

I don't think this negates the fact that this study is interesting and useful, but that the claims must be tempered in light of this information (i.e., the fact that O-H cannot be readily detected needs to be disclosed).

Response: We thank the reviewer for the helpful comments. We understand the reviewer's concern on the ability to detect O-H species and thus the applicability of the approach on other metal oxides. As requested by the reviewer, we performed additional NMR experiments to show that bare surface sites and hydroxyl species can be observed at the same time (Supplementary Figure 10 and 11).

The ^{17}O NMR spectrum of ^{17}O enriched NS-773 (enriched in $^{17}\text{O}_2$) exposed to H_2^{17}O (90 %, 4 mbar) shows a new broad peak centered at -100 ppm with a quadrupolar line shape, in addition to the high frequency peaks at 47 to 39 ppm due to bare surface sites (Supplementary Figure 10). A simulation on the line shape of the broad peak gives the following NMR parameters for this site ($C_Q = 7.6$ MHz, $\eta = 0.1$, $\delta_{\text{iso}} = 5$ ppm) (dashed line Supplementary Figure 10). These NMR parameters are quite similar to hydroxyls in $\text{Mg}(\text{OH})_2$ ($C_Q = 6.8$ MHz, $\eta = 0$ and $\delta_{\text{iso}} = 25$ ppm, ref. J. Magn. Reson. 1988, 76, 106.), and therefore this signal can be assigned accordingly.

We have also tried another way to enrich MgO nanosheets. After exposing non-enriched NS-773 to H_2^{17}O (90 % ^{17}O , 16 mbar), the ^{17}O NMR spectrum of the resulting solid shows a very broad peak with small and relatively sharp components at higher frequencies (47 to 39 ppm) owing to bare surface sites (bottom of Supplementary Figure 11). Again the broad resonance can be assigned to surface hydroxyl like species. After heating this sample to 589 K under vacuum, the decomposition temperature of $\text{Mg}(\text{OH})_2$ {ref. J. Green, J. Mater. Sci., 1983, 18, 637}, the high frequency peaks at 47 to 39 ppm becomes much stronger while the broad signal decreases significantly in intensity in the ^{17}O NMR spectrum, indicating that dehydroxylation occurs and a large fraction of hydroxyl species have been converted to bare surface sites (top of Supplementary Figure 11). This result shows that MgO nanosheets can be enriched by using H_2^{17}O . We did not heat the sample to higher temperatures (e.g., 773 K) because that would cause most of the ^{17}O ions to diffuse to the bulk part of the sample.

Therefore, we are able to observe bare oxygen sites (sharp components at high frequency of 47 to 39 ppm) and hydroxyl species (low frequency broad peak) at the same time with our method, however, these samples have much higher concentrations of hydroxyl sites than NS-773. We do agree with the reviewer that the sites with large C_Q (e.g., OH) are challenging to observe at low concentrations, which is also the case here for hydroxyl species and explains the absence of the corresponding peak in dry samples. Therefore, we have added related discussion on the limitations of our approach at the end of the “Discussion” section. Now it reads (the method can) “potentially be extended to a variety of other metal oxides to investigate gas adsorption processes, and help design related materials with improved properties for adsorption and catalysis. Despite these possible promising applications, it may be challenging to detect species at a very low concentration with a large quadrupolar interaction by using this approach. Although oxygen ions in most metal oxides do have a small to medium C_Q due to the small electronegativity values of common metals{ref. Schramm and Oldfield, J. Am. Chem. Soc., 1984, 106, 2502}, large quadrupolar interactions may be found in oxygen atoms bound to elements with a relatively large electronegativity, such as hydroxyl groups in this case, or in surface sites with lower coordination numbers and/or dangling bonds{ref. M. Wang et al., Sci. Adv., 2015, 1, e1400133}. Obtaining data at a higher external magnetic field, which decreases the line broadening due to quadrupolar interactions, may alleviate the problem.”